# Safety and Efficacy of *Bojungikki-Tang* in Advanced NSCLC Patients Receiving Treatment with Immune Checkpoint Inhibitors: Protocol for a Multicenter, Double-Blind, Randomized, Placebo-Controlled Pilot Trial

**DOI:** 10.3390/ijerph20054507

**Published:** 2023-03-03

**Authors:** Mi Mi Ko, Mi-Kyung Jeong, Chang Min Choi, Seung Hyeun Lee, Jaemoo Chun, Jin-Mu Yi, Ho Jang, Sung Yong Lee

**Affiliations:** 1KM Science Research Division, Korea Institute of Oriental Medicine, Daejeon 34054, Republic of Korea; 2KM Convergence Research Division, Korea Institute of Oriental Medicine, Daejeon 34054, Republic of Korea; 3Department of Pulmonary and Critical Care Medicine, Asan Medical Center, College of Medicine, University of Ulsan, Seoul 05505, Republic of Korea; 4Division of Pulmonary and Critical Care Medicine, Department of Internal Medicine, Kyung Hee University Medical Center, Kyung Hee University College of Medicine, Seoul 02447, Republic of Korea; 5KM Data Division, Korea Institute of Oriental Medicine, Daejeon 34054, Republic of Korea; 6Division of Pulmonary, Allergy, and Critical Care Medicine, Department of Internal Medicine, Korea University Guro Hospital, Korea University College of Medicine, Seoul 08308, Republic of Korea

**Keywords:** *Bojungikki-tang*, immune checkpoint inhibitors, non-small cell lung cancer

## Abstract

Cancer immunotherapy with immune checkpoint inhibitors (ICIs) is a major treatment option for several types of cancer, including non-small cell lung cancer (NSCLC). The proposed study aims to investigate the safety and efficacy of *Bojungikki-tang* (BJIKT) therapy (an herbal medicine) in patients with advanced NSCLC treated with ICIs. This multicenter, randomized, placebo-controlled pilot study will be performed at three academic hospitals. Thirty patients with advanced NSCLC, undergoing atezolizumab monotherapy as second- and subsequent-line treatment, will be recruited and randomly assigned to either BJIKT treatment (atezolizumab + BJIKT) or placebo (atezolizumab + placebo). The primary and secondary outcomes are the incidence of adverse events (AEs), including immune- related AEs (irAEs) and non-immune-related AEs (non-irAEs); and early termination rate, withdrawal period, symptom improvement of fatigue, and skeletal muscle loss, respectively. The exploratory outcomes are patient objective response rate and immune profile. This is an ongoing trial. Recruitment started on 25 March 2022 and is expected to be completed by 30 June 2023. This study will provide basic evidence for the safety profiles, including irAEs, of herbal medicine in patients with advanced NSCLC treated with ICIs.

## 1. Introduction

Lung cancer causes the greatest number of deaths among all cancers worldwide, and the five-year survival rate for patients diagnosed with lung cancer is approximately 18% to 27% [1,2]. Non-small cell lung cancer (NSCLC), which accounts for 85% of lung cancers, is most frequently treated with immunotherapy [3]. Notably, immune checkpoint inhibitors (ICIs) have emerged as a major treatment for several types of cancer, including NSCLC.

ICIs, which are monoclonal antibodies targeting programmed cell death receptor 1 (PD-1) and programmed cell death ligand 1 (PD-L1), inhibit immune evasion mechanisms by blocking the PD-1/PD-L1 signaling pathway and reinvigorating T-cell activation. ICIs include anti-PD-1 antibodies, such as pembrolizumab and nivolumab; and anti-PD-L1 antibodies, such as atezolizumab. These medicines are used as standard treatments for advanced or metastatic NSCLC patients with no driver oncogenes (e.g., epidermal growth factor receptor (EGFR) and anaplastic lymphoma kinase (ALK)) [4].

However, despite their important clinical benefits, ICIs are associated with a unique spectrum of adverse reactions, termed immune-related adverse events (irAEs).

These irAEs include diarrhea/colitis, fatigue, nausea/vomiting, rash, and pruritus and may necessitate treatment discontinuation [5,6,7,8,9,10,11]. In a previous study of 61 NSCLC patients receiving ICIs, the discontinuation rate due to non-irAEs was 60%, and that due to irAEs reached 30% [5].

Moreover, when ICIs are used in combination with chemotherapy, the incidence of serious adverse events (AEs) of all grades and AEs above grade 3 were greater than for treatment with ICIs alone [6]. In addition, the combination of two ICIs or the combination of ICIs with cytotoxic chemotherapy is associated with higher rates of toxicity and increased severity of irAEs or treatment-related AEs [7,8]. Fatigue is one of the most common AEs of ICI treatment in lung cancer patients [6,9]. This symptom frequently occurs when atezolizumab monotherapy is administered to patients with advanced lung cancer (27%) [10].

*Bojungikki-tang* (BJIKT), also known as “*Bu-Zhong-Yi-Qi-Tang*” in China and “*Hochuekki-to*” in Japan, is one of the most commonly used herbal medicines for the treatment of various symptoms caused by gastric disorders (including fatigue, weariness, physical weakness resulting from illness, anorexia, and night sweats) in East Asian countries [12,13]. This herbal medicine is composed of *Ginseng Radix, Atractylodis Rhizoma Alba, Astragali Radix, Angelicae Gigantis Radix, Zizyphi Fructus, Bupleuri root, Citri Unshius Pericarpium, Cimicifugae Rhizoma, Zingiberis Rhizoma Recens, and Glycyrrhizae Radix et Rhizoma*. It has a bidirectional immunomodulatory effect that compensates for both excess and deficient conditions in treating cancer [14,15,16,17,18,19,20]. Many studies have reported on the safety of BJIKT, including its toxicity profile and long-term administration in cancer patients [21,22,23,24,25,26,27,28,29]. However, to our knowledge, no previous study has evaluated the irAEs of BJIKT in cancer patients treated with ICIs.

Therefore, in this protocol, we propose a randomized, placebo-controlled, double- blinded trial, to investigate the safety and efficacy for BJIKT in patients with advanced NSCLC who are concomitantly being treated with ICIs. Overall, this trial aims to: (1) investigate the safety of BJIKT in advanced NSCLC patients treated with ICIs in comparison with a placebo-controlled group; (2) evaluate post-treatment symptom improvement, focusing on fatigue and skeletal muscle loss in BJIKT and placebo-controlled groups; and (3) explore the feasibility of an integrative cancer treatment strategy by examining the patient immune status and objective response rate (ORR) following combined ICI and BJIKT treatment.

## 2. Materials and Methods

### 2.1. Study Design and Setting

This multicenter, randomized, placebo-controlled pilot study will be performed at Korea University Guro Hospital, Asan Medical Center, and Kyung Hee University Hospital (all located in Seoul, Republic of Korea). A total of 30 patients with NSCLC scheduled to undergo second- and subsequent-line monotherapy with atezolizumab, an ICI, will be recruited after obtaining their written informed consent. The enrolled participants will be randomly allocated into two parallel groups: BJIKT (atezolizumab + BJIKT) and placebo (atezolizumab + placebo). Patients will receive intravenous atezolizumab (at a 1200 mg fixed dose) every three weeks on day 1 of each three-week cycle [30]. According to the allocated group, patients will take BJIKT or placebo (4 g at a time) every day, regardless of withdrawal of the base drug, starting from the initiation of atezolizumab therapy. The trial will consist of three cycles administered over the course of approximately nine weeks, with a screening period and three study visits at three-week intervals. All participants will undergo the necessary examinations and testing procedures within four weeks prior to enrollment (i.e., prior to administering the investigational drug) according to the described clinical trial protocol.

The study period procedure is summarized in the Consolidated Standards of Reporting Trials (CONSORT) diagram (Figure 1), and this protocol is written in accordance with the recommendations of the Standard Protocol Items: Recommendations for Interventional Trials (SPIRIT) checklist. The schedule of enrolment, intervention, and assessments is summarized in Table 1.

### 2.2. Recruitment

Recruitment for this investigation began in March 2022 and is expected to end by June 2023. All participants will undergo a standardized interview and will receive clinical study information about the trial. Written consent will be obtained from all participants. The purpose, procedures, potential risks, and benefits of the study will also be thoroughly explained to the participants. The participants will be able to withdraw from the study at any time without any adverse consequences.

### 2.3. Participants

#### 2.3.1. Inclusion Criteria

Patients who meet all of the following criteria will be included in this trial: (1) adult male and female patients aged ≥19 years; (2) patients diagnosed with stage IIIB, IIIC, or IV NSCLC according to the TNM 8th edition diagnostic criteria; (3) patients scheduled to receive monotherapy with atezolizumab, an ICI, as second- and subsequent-line therapy (after providing their informed consent); (4) patients with advanced or metastatic NSCLC whose disease has progressed during or after approved platinum-based chemotherapy; (5) patients who are expected to survive for more than three months; (6) patients with an Eastern Cooperative Oncology Group (ECOG) performance status (PS) score of ≤2; (7) patients with measurable or evaluable lesions, as defined in the Response Evaluation Criteria in Solid Tumors (RECIST) version 1.1; (8) a hematological function evaluation showing a hemoglobin level of ≥9.0 g/dL, an absolute neutrophil count of ≥1500/µL, and a platelet count of ≥100 × 10^3^/µL; (9) renal function with a creatinine level of ≤1.5× the upper limit of normal (ULN), and creatinine clearance of ≥45 mL/min; (10) a liver function evaluation showing alanine aminotransferase (ALT) and aspartate aminotransferase (AST) levels ≤ 2.5× ULN; and (11) patients who voluntarily decide to participate and consent to study participation in writing after listening to and understanding a detailed explanation of the clinical trial.

#### 2.3.2. Exclusion Criteria

The study exclusion criteria are as follows: (1) diagnosis of a new primary cancer within the last five years; (2) treatment with an immunosuppressive drug within the last two weeks or ICI treatment within the last six weeks; (3) treatment with antibiotics within the last two weeks; (4) current treatment with thiazide or a loop diuretic; (5) the presence of hypokalemia; (6) a history of interstitial lung disease in the patient’s medical history; (7) confirmed EGFR or ALK gene mutations; (8) active autoimmune disease requiring systemic therapy within the last two years; (9) uncontrolled diabetes (i.e., diabetes that is uncontrolled with insulin or oral medication, based on fasting blood sugar levels); (10) uncontrolled hypertension; (11) heart defect; (12) active or untreated human immunodeficiency virus, tuberculosis, hepatitis B, or hepatitis C; (13) significant hematologic disease within six months before randomization; (14) abdominal cavity, esophageal atresia, or gastrointestinal-duct perforation within six months before randomization; (15) unresolved toxicity with a Common Terminology Criteria for Adverse Events (CTCAE) v5.0 level of >1 resulting from previous therapy at the start of the trial therapy; (16) serious hypersensitivity to ingredients in the investigational drug; (17) pregnancy or lactation; (18) among fertile women, lack of consent to use effective contraception for at least five months after the final administration of atezolizumab during the course of the clinical trial; (19) use of an oriental medicine or functional food that was determined by the investigator as having a potential to affect the trial or the patient’s safety within four weeks before initial treatment with the investigational drug; (20) participation in a clinical trial within four weeks before the first administration of the investigational drug; and (21) patients deemed unsuitable for this clinical trial by the investigator for various reasons, including serious infectious disease and organ failure.

### 2.4. Randomization and Allocation Concealment

A total of 30 eligible patients will be randomized into one of the two groups in a 1:1 ratio. An independent medical statistician will generate a random assignment code using the block randomization method with block sizes in SAS^®^ Version 9.4 (SAS Institute Inc., Cary, NC, USA). Each participant who satisfies the eligibility criteria will be assigned to the BJIKT treatment or placebo group at the baseline visit with a 1:1 allocation ratio.

The unique code derived for the random number table for each participant will be placed in an opaque sealed envelope and stored in double-locked cabinets. Labeling will be similar for both groups. We will maintain allocation concealment throughout the study period by providing participants with identically packaged, consecutively numbered drug containers.

### 2.5. Blinding

The proposed study will be a double-blinded trial wherein both participants and researchers, including nurses, physicians, and statisticians, will be blinded to the treatment allocation. Only an independent statistician will be aware of the group allocation and will have access to the randomization list and blinding codes.

### 2.6. Interventions

The BJIKT will be prepared by Hanpoong Pharm & Foods Co., Ltd. (Iksan, Republic of Korea). It will be produced according to the Korean Good Manufacturing Practice guidelines outlined by Korea Ministry of Food and Drug Safety (MFDS) and manufactured so that there is no difference between them in shape, color, smell, or taste. A blinded pharmacist will check the storage, dispensation, and quality of the investigational products. 

The BJIKT used in this study will be a dried, light yellowish-brown to fine brown granule extracted with water. The extract is composed of 10 medicinal plants, as follows: *Ginseng Radix* 1.33 g, Atractylodis Rhizoma Alba 1.33 g, Astragali Radix 1.33 g, *Angelicae Gigantis Radix* 1.0 g, *Zizyphi Fructus* 0.67 g, *Bupleuri* root 0.67 g, *Citri Unshius Pericarpium* 0.67 g, *Cimicifugae Rhizoma* 0.33 g, *Zingiberis Rhizoma Recens* 0.17 g, and *Glycyrrhizae Radix et Rhizoma* 0.5 g (representing the dose for one day of treatment).

The placebo will be made of water-immersed crude extract, lactose hydrate, corn starch, hydroxypropyl cellulose, sucrose fatty acid ester, magnesium stearate, and light anhydrous silicic acid. Both drugs will be identical in appearance, shape, taste, and color (brown). The BJIKT and placebo granules will be sealed in opaque aluminum bags and will be administered to the participants in 4g doses. 

All enrolled participants will receive intravenous atezolizumab (1200 mg fixed dose) every three weeks on day 1 of each three-week cycle and will take BJIKT or placebo (4 g at a time) every day for approximately nine weeks regardless of withdrawal of the base drug, starting from the initiation of atezolizumab therapy. This regimen may be discontinued according to the investigator’s judgment. All participants will be requested to return the remaining drugs in order to calculate compliance. All medications and treatments that are clinically determined to be necessary for the patient can be prescribed. The investigators will record the details of concomitant medication and treatment information in the case report form during this clinical trial period (from the date of clinical trial registration to the end of observation for each subject).

### 2.7. Outcome Measures

#### 2.7.1. Primary Outcomes

The primary study outcomes are the incidence of AEs, including irAEs and non- irAEs, and the incidence of AEs of CTCAE grade 4 or higher. These will be graded using the National Cancer Institute CTCAE v4.03 criteria [31].

#### 2.7.2. Secondary Outcomes

The secondary outcomes of this study include the early interruption rate; the withdrawal period (including the early interruption rate due to AEs of CTCAE grade 3 or above); the incidence of adverse drug reactions (ADRs, including irAEs for atezolizumab monotherapy and combined therapy [30,32,33] at CTCAE grade 3 or above with an incidence rate of over 1%); fatigue symptom improvement using the Functional Assessment of Chronic Illness Therapy—Fatigue domain (FACIT-F; Trial Outcome Index (TOI)—Fatigue) [34], the European Organization for the Research and Treatment of Cancer Quality of Life Questionnaire (EORTC-QLQ-C30) fatigue subscale [35,36,37], and the numeric rating scale (NRS) for fatigue; evaluations of skeletal muscle loss (using the strength, assistance with walking, rising from a chair, climbing stairs, and falls (SARC-F) questionnaire [38], a body composition test, the sit-to-stand test, and the thoracic muscle index), changes in the ECOG PS, pattern identification diagnosis, and treatment compliance. Demographic information such as age, sex, medical history, and laboratory tests will also be recorded.

Early discontinuation will be defined as failure to complete the planned study administration protocol regardless of the cause or dosing regimen, except when discontinuation is due to progressive disease. In addition, even if the planned three-cycle schedule has been completed, if the actual number of ICI doses during the treatment period is less than 60% of the total planned number due to toxicity, these patients will be included in the category of early discontinuation. To investigate therapeutic effects based on pattern identification, the cold–heat pattern identification questionnaire will be used to differentiate cold, non-cold, heat, and non-heat type classifications. This questionnaire consists of a five-point scale with 15 questions evaluating cold and heat types [39,40].

#### 2.7.3. Exploratory Outcomes

The exploratory outcomes for this study are the ORR (defined as the proportion of patients with the best overall complete or partial response) of atezolizumab treatment according to the RECIST version 1.1 guidelines and immune profiles.

Blood samples (approximately 12 mL) will be collected for immune profiling analysis at the time points of pre-treatment and after treatment of 3 cycles (EOT). Additionally, sample collection at first cycle treatment will proceed according to the investigator’s judgement. Basic immune cell subtypes, fractions of Th1/Th2/Th17, Treg subtypes, naïve/memory/senescence markers, and the expression of immune checkpoint receptors will be measured through a multicolor fluorescence-activated cell sorting analysis of peripheral blood mononuclear cell (PBMC) samples. Cytokine levels will be measured using a multiplex assay. Additionally, Th1, Th2, and Th17 cell fractions and natural killer cell activity will be measured. Moreover, a total transcriptome analysis of PBMCs will be performed via sequencing.

#### 2.7.4. Safety Outcomes

At each visit, all adverse reactions reported by the participant and observed by the investigator will be recorded, including those pertaining to vital signs and blood tests. All AEs (including irAEs) and ADRs related to ICI treatment in NSCLC patients will be descriptively recorded with detailed explanations, including information about the seriousness, severity, and causality of the events. Safety and tolerability will be assessed according to the National Cancer Institute Common Terminology Criteria (CTCAE v4.03). If possible, AEs collected during the trial period will be recorded using Medical Dictionary for Regulatory Activities (MedDRA) terms through a case report form.

### 2.8. Sample Size

To our knowledge, this is the first exploratory clinical trial to investigate the improvement in symptoms and AEs following BJIKT treatment as compared to placebo in patients with advanced NSCLC who are being treated with ICIs.

Considering the exploratory nature of the clinical trial and the number of patients available from each center, a total of 30 participants (15 in each group) will be enrolled. Recruitment started on 25 March 2022 and is expected to be completed by 30 June 2023.

### 2.9. Statistical Analysis

All statistical analyses will be performed using a two-tailed test with a 5% level of significance. An independent statistician will conduct statistical analysis using SAS^®^ Version 9.4 (SAS Institute Inc., Cary, NC, USA). 

The primary efficacy variable will be summarized using the full analysis set (FAS), and the per-protocol set (PPS) will be used for a supportive analysis. All other efficacy analyses will be based on the FAS. Safety variables will be summarized using a safety set (SS), which includes all participants receiving at least one dose of the study medication. The FAS will consist of all randomized participants who received at least one dose of the study medication and who underwent a valid measurement of the primary efficacy variable at baseline.

The PPS will consist of all participants in the FAS who showed no important protocol deviation affecting the primary efficacy variable.

Participants showing important protocol deviations will be predefined and evaluated during a data evaluation meeting prior to unblinding the data.

Categorical data will be presented as frequencies and percentages, and continuous data will be presented as means and 95% confidence intervals. 

The normality of continuous variables in the clinical data will be tested using the Kolmogorov–Smirnov test. Continuous data will be analyzed using the independent t-test or Wilcoxon rank-sum test, and categorical data will be analyzed using the chi-squared test or Fisher’s exact test for comparison of two groups. Mixed-effect model repeated measures (MMRM) will be used, with each group and visit time as fixed factors and subjects as random factors. Paired *t*-tests or Wilcoxon signed rank tests will be used to compare the factors before and after treatment. 

Pearson’s correlation or Spearman’s rank correlation tests will be used for each administration group in order to examine the correlations among various immune indicators measured in blood samples, treatment responses, and each identified pattern. The data collected in this study will be used as is for analysis without replacing or imputing missing values. 

### 2.10. Data Management and Monitoring

Clinical trial monitoring will be conducted through regular site visits and telephone calls by a clinical research associate (CRA) on the monitoring committee in order to monitor protocol compliance, the recruitment rate, document reporting, and AEs during the trial period. If any problems arise during the clinical trial process, the CRA will consult with the investigator. 

The trial data will be saved in an electronic data capture system (Medidata Rave; Medidata Solutions Inc., New York, NY, USA) used for data collection and verification. The trial data will be managed by an independent data management (DM) team. The electronic CRF is accessible only to investigators who are directly involved in the clinical trial and receive relevant education. The range of the data values will be set in advance to ensure the data quality, and data entry will be performed only once by the investigators at the affiliated institutions. Data quality will be managed through two validation processes by the CRA and DM team. All data and documents obtained from the clinical trial will be preserved in a separate storage location. No information will be released without the permission of the principal investigator, and all data acquired during the trial will be stored separately in accordance with institutional review board (IRB) regulations.

### 2.11. Ethics and Dissemination

This study was approved by the IRB of Korea University Guro Hospital (approval number: 2021GR0472), Asan Medical Center (approval number: S2021-2096-0001), and Kyung Hee University Hospital (approval number: KHUH2021-08-079-001) on 30 September 2021. The trial was registered with the Clinical Research Information Service (https//cris.nih.go.kr/cris; identifier number, KCT0006689, accessed on 28 October 2021) in October 2021. This study is based on version 5.2 of the protocol. This is an ongoing trial. Recruitment started on 25 March 2022 and will be expected to be completed by 30 June 2023. 

All patient participation will be considered voluntary, and patients will be informed in advance that they can withdraw from the study at any time without penalty or adverse consequences. 

## 3. Discussion

Herein, we reported a study protocol for a pilot multicenter, randomized, double- blinded, placebo-controlled study assessing the safety and efficacy of BJIKT in advanced NSCLC patients concomitantly treated with ICIs. In addition, we will evaluate the feasibility of an integrative cancer treatment strategy by examining immune profiles and treatment responses to combined treatment with ICIs and herbal medicine.

ICIs have emerged as front-line therapeutic option for advanced NSCLC [4]. However, ICIs are reported to be associated with several AEs, including irAEs, that affect treatment discontinuation [5,6,7,8,9,10,11]. Fatigue is one of the most common AEs associated with ICI treatment in patients with lung cancer [6,9,10]. Furthermore, fatigue is a common symptom in advanced NSCLC, and patients with stage IV disease experience fatigue and weight loss at a higher rate than those at lower stages; thus, such symptoms and progression are more likely to occur in advanced-stage lung cancer [41]. Moreover, cancer-related fatigue (CRF) and loss of skeletal muscle in cancer patients are known to result from systemic inflammation [7,42,43,44]. However, maintenance of skeletal muscle through exercise is the only solution to this, as no active treatment plan has been proposed thus far [45,46,47]. In this regard, BJIKT can be considered a treatment alternative for the symptoms of fatigue and decreased skeletal muscle mass in cancer patients.

BJIKT is an herbal formula approved by the Korean MFDS as a medicine for weak constitution, tiredness, weakness after illness, anorexia, and night sweating. BJIKT exerts beneficial effects on CRF and quality of life in patients with cancer [13]. 

Moreover, BJIKT treatment has been shown to inhibit the production of proinflammatory cytokines and is considered a beneficial treatment for restoring the immunological status of cancer patients experiencing physical and psychological stress [14,15,16,17]. In addition, BJIKT enhances weakened immune function and suppresses the enhanced immune function attributable to the inflammatory response, suggesting that this medicine has a bidirectional immunomodulatory effect that compensates for both excess and deficiency conditions [18,19,20]. Although several studies have reported on the safety of BJIKT [21,22,23,24,25,26,27,28,29], none has evaluated the irAEs of BJIKT in patients with advanced NSCLC treated with ICIs.

The findings of our study will provide basic evidence regarding the safety and efficacy of BJIKT in the treatment of patients with advanced NSCLC who are being treated with ICIs through a rigorously well-designed clinical trial. In addition, we will explore the therapeutic effects based on pattern identification, the most important factor for diagnosis and treatment in clinical fields [48,49,50,51,52].

However, this clinical trial is limited in that the survival benefits of combination therapy will not be evaluated due to the short trial period, and it will be conducted with a small sample size, so further clinical trials with large sample sizes need to be conducted to confirm safety and therapeutic effect of the coadministration of ICIs and BJIKT.

## 4. Conclusions

This trial will be the first study to evaluate the safety and feasibility for ICIs and BJIKT as an integrative cancer treatment in advanced NSCLC patients. This study will provide basic evidence for the safety profiles, including irAEs, of herbal medicine in patients with advanced NSCLC who are being treated with ICIs. Hopefully, the findings of our study will provide basic evidence for further integrative research aimed at the survival benefits of combined treatment with ICIs and herbal medicine.

## Figures and Tables

**Figure 1 ijerph-20-04507-f001:**
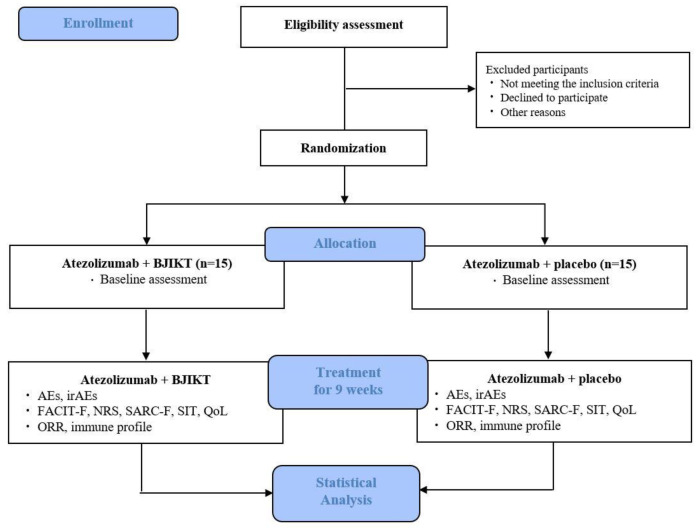
Study flowchart. BJIKT, *Bojungikki-tang*; AE, adverse event; irAEs, immune-related adverse events; FACIT-F, Functional Assessment of Chronic Illness Therapy for Fatigue; NRS, Numeric Rating Scale; SARC-F, strength, assistance with walking, rising from a chair, climbing stairs, falls; SIT, sit-to-stand; QoL, quality of life; ORR, objective response rate.

**Table 1 ijerph-20-04507-t001:** Schedule of enrollment, interventions, and assessments.

	Study Period
	Screening	Treatment & F/U
	Baseline	V1	V2	V3 ^c^
Time Point	~4 Weeks	C1D1(±7 Days)	C2D1(±7 Days)	C3D1(±7 Days)	C3D21(±7 Days)
**Enrollment**					
Informed consent	X				
Eligibility screen	X	X			
Demographic characteristics	X				
Medical history	X	X			
PD-L1 testing(immunohistochemistry), EGFR/ALK mutation test	X				
Physical examination and vital signs	X	X	X		X
Electrocardiography	X				
ECOG PS	X	X	X		X
Random allocation		X			
**Interventions**					
Arm 1: atezolizumab + BJIKT		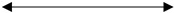	
Arm 2: atezolizumab + placebo		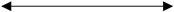	
**Assessments**					
EORTC-QLQ-C30	X		X		X
Fatigue subscale/NRS	X		X		X
FACIT-F(TOI-F)	X		X		X
SARC-F	X		X		X
Muscle test (InBody^TM^, STS test)	X		X		X
Cold/heat pattern	X		X		X
Laboratory testing ^a^	X	^b^	X		X
Immunological testing (blood collection)	X	^b^			X
Chest CT	X				X
X-ray examination		X	X	X	
Body composition test ^d^		X	X	X	X
Tumor evaluation (according to RECIST v1.1)	X				X
Adverse events (CTCAE v4.03)		X	X	X	X
Medication compliance			X	X	X

^a^ Hematological, blood biochemical, CRP, urine, and pregnancy testing. ^b^ Execute if necessary, according to the investigator’s judgment. ^c^ End of treatment (EOT). ^d^ Body weight, body fat percentage, body fat mass, skeletal muscle mass; ALK, anaplastic lymphoma kinase; BJIKT, *Bojungikki-tang*; C, cycle; CRP, C-reactive protein; CT, computed tomography; CTCAE, Common Terminology Criteria for Adverse Events; D, day(s); ECOG PS, Eastern Cooperative Oncology Group Performance Status; EGFR, epidermal growth factor receptor; EORTC-QLQ-C30, European Organization for Research and Treatment of Cancer Quality of Life Questionnaire; FACIT-F(TOI-F), Functional Assessment of Chronic Illness Therapy for Fatigue; F/U, follow-up; NRS, Numeric Rating Scale; RECIST, Response Evaluation Criteria in Solid Tumors; STS, sit-to-stand; SARC-F, strength, assistance with walking, rising from a chair, climbing stairs, and falls questionnaire.

## Data Availability

The data presented in this study are available on request from the corresponding author.

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
