# Peer review of "Safety and Efficacy of *Bojungikki-Tang* in Advanced NSCLC Patients Receiving Treatment with Immune Checkpoint Inhibitors: Protocol for a Multicenter, Double-Blind, Randomized, Placebo-Controlled Pilot Trial"

_ijerph, 2023, doi:10.3390/ijerph20054507_

Round 1
Reviewer 1 Report
I think the paper will be really interesting if contain results. Now in this form only a simple description of the materials and methods. So I think in this form the paper not suitable for publication.
Author Response
Reviewer 1
Point 1
I think the paper will be really interesting if contain results. Now in this form only a simple description of the materials and methods. So I think in this form the paper not suitable for publication.
Response 1: Thank you for your comment. we acknowledge that this manuscript mainly provides a description of the materials and methods as protocol, and does not contain any results yet. However, we believe that publishing the protocol is important to promote transparency and prevent selective outcome reporting. This study is designed as an exploratory trial to investigate the safety and efficacy of combined treatment with herbal medicine and immune checkpoint inhibitor (ICI). Once the study is completed, we will report the findings from this study. We expect that our study will provide basic evidence for futher integrative research aimed to survival benefits of combined treatment with ICIs and herbal medicine.
Reviewer 2 Report
This manuscript outlines a protocol to conduct a randomized, placebo-controlled, double-blinded trial, to investigate the safety and efficacy for BJIKT in patients with advanced NSCLC receiving immune checkpoint inhibitor (atezolizumab). This study is well rationalized and designed.
Minor comments:
-The authors should clarify when and how many times they intend to collect blood samples in the exploratory outcome section. Collecting blood at mid-point of the treatment cycle may also be beneficial.
-Statistical analysis section should include what is the minimum number of patients required in each group to ascertain significance.
-Do the authors have a contingency plan in case they are unable to recruit the required number of patients?
Author Response
Reviewer 2
This manuscript outlines a protocol to conduct a randomized, placebo-controlled, double-blinded trial, to investigate the safety and efficacy for BJIKT in patients with advanced NSCLC receiving immune checkpoint inhibitor (atezolizumab). This study is well rationalized and designed.
Minor comments:
Point 1
-The authors should clarify when and how many times they intend to collect blood samples in the exploratory outcome section. Collecting blood at mid-point of the treatment cycle may also be beneficial.
Response 1: Thank you for your comment. As your comment, we revised methods of collecting blood samples to be more explicit in the exploratory outcome section.
“~ Blood samples (approximately 12 mL) will be collected for immune profiling analysis at the time points of pre-treatment and after treatment of 3 cycles (EOT). Also, sample collection at first cycle treatment will be proceeded according to the investigator’s judgement.~”
Point 2
-Statistical analysis section should include what is the minimum number of patients required in each group to ascertain significance.
Response 2: Thank you for your comment. We agree with your comment. However, this is the first exploratory clinical trial to investigate the improvement in symptoms as well as AEs following BJIKT treatment as compared to placebo in patients with advanced NSCLC who are being treated with ICIs. As mentioned, this is a pilot study and we could not find adequate references regarding the samlpe size. Hence, although the number of recruited patients is limited, considering the exploratory nature of the clinical trial and the number of patients available from each center, a total of 30 participants (15 in each group) will be enrolled.
Point 3
-Do the authors have a contingency plan in case they are unable to recruit the required number of patients?
Response 3: Thank you for your comment. As mentioned, considering the exploratory nature of the clinical trial and the number of patients available from each center, a total of 30 participants (15 in each group) will be enrolled. We can expect that recruitment for this investigation will be ended to be completed by June 2023 considering the current enrolled situation of each center.
Reviewer 3 Report
I have no comments. The authors describe the protocol for the Safety and Efficacy of Bojungikki-Tang in Advanced NSCLS Patients, however, the study is still in progress and there are no conclusive results. The manuscript discussion and conclusion are only descriptive. No decisions can be made on this manuscript at the present time, it can be reconsidered only after the completion of the study.
Author Response
Reviewer 3
Point 1
I have no comments. The authors describe the protocol for the Safety and Efficacy of Bojungikki-Tang in Advanced NSCLS Patients, however, the study is still in progress and there are no conclusive results. The manuscript discussion and conclusion are only descriptive. No decisions can be made on this manuscript at the present time, it can be reconsidered only after the completion of the study.
Response 1: We thank the reviewer for pointing out this. This manuscript describes the protocol of exploratory clinical trial aimed to investigate the improvement of symptoms and the possible onset of adverse events in the immunotherapy of non-small cell lung cancer with addition of the herbal medicine Bojungikki-tang commonly used in Asian countries including Korea. We Know that publishing a protocol before the completion of the trial is considered positively, it would help improve the transparency of the research.
This trial is expected to conclude on June 30, 2023. We will report the findings from this study. This study will provide basic evidence for the safety profiles, including irAEs of herbal medicine in patients with advanced NSCLC who were treated with ICIs. Hopefully, the findings of our study will provide basic evidence for further integrative research aimed to survival benefits of combined treatment with ICIs and herbal medicine.
Reviewer 4 Report
Considering the interest of the readers, I suggest the speedy publication of the results obtained from the pilot study, which would justify the subsequent, multicentric development of the research.
Author Response
Reviewer 4
Point 1
Considering the interest of the readers, I suggest the speedy publication of the results obtained from the pilot study, which would justify the subsequent, multicentric development of the research.
Response 1: Thank you for your comment. This trial is expected to conclude on June 30, 2023. We will report the findings from this study. This study will provide basic evidence for the safety profiles, including irAEs of herbal medicine in patients with advanced NSCLC who were treated with ICIs. Hopefully, the findings of our study will provide basic evidence for further integrative research aimed to survival benefits of combined treatment with ICIs and herbal medicine.
Round 2
Reviewer 1 Report
Dear Authors
as i mentioned in my previouse review i think the paper is intresting i hope you will get a results soon and you will able to present it too.
Anyway i ahve checked teh simillar papers and i think as protocoll is suitable for publishig